# Evaluation of a Lateral Flow Immunoassay for Rapid Detection of CTX-M Producers from Blood Cultures

**DOI:** 10.3390/microorganisms11010128

**Published:** 2023-01-04

**Authors:** Hanshu Fang, Chung-Ho Lee, Huiluo Cao, Shuo Jiang, Simon Yung-Chun So, Cindy Wing-Sze Tse, Vincent Chi-Chung Cheng, Pak-Leung Ho

**Affiliations:** 1Department of Microbiology, Carol Yu Centre for Infection, University of Hong Kong, Hong Kong, China; 2Department of Clinical Pathology, Kwong Wah Hospital, Hospital Authority, Hong Kong, China; 3Department of Microbiology, Queen Mary Hospital, Hospital Authority, Hong Kong, China

**Keywords:** cefotaxime resistance, beta-lactamases, CTX-M, extended-spectrum beta-lactamases, immunochromatographic assay, *Enterobacteriaceae*, bacteremia, blood-stream infection, blood culture

## Abstract

Bacteremia caused by extended-spectrum β-lactamases-producing *Enterobacterales* has increased rapidly and is mainly attributed to CTX-M enzymes. This study aimed to evaluate the NG-Test^®^ CTX-M MULTI lateral flow assay (CTX-M LFA) for rapid detection of CTX-M producers in blood cultures (BCs) positive for Gram-negative bacilli in spiked and clinical BCs. Retrospective testing was performed on BC bottles spiked with a collection of well-characterized *Enterobacterales* isolates producing CTX-M (*n* = 15) and CTX-M-like (*n* = 27) β-lactamases. Prospective testing of clinical, non-duplicate BCs (*n* = 350) was performed in two hospital microbiology laboratories from April 2021 to March 2022 following detection of Gram-negative bacilli by microscopic examination. Results were compared against molecular testing as the reference. In the spiked BCs, the CTX-M LFA correctly detected all CTX-M producers including 5 isolates with hybrid CTX-M variants. However, false-positive results were observed for several CTX-M-like β-lactamases, including OXY-1-3, OXY-2-8, OXY-5-3, FONA-8, -9, -10, 11, 13 and SFO-1. In clinical BCs, the CTX-M LFA showed 100% (95% CI, 96.0–100%) sensitivity and 99.6% (97.9–100%) specificity. In conclusion, this study showed that rapid detection of CTX-M producers in BC broths can be reliably achieved using the CTX-M LFA, thus providing an opportunity for early optimization of antibiotics.

## 1. Introduction

The prevalence of bloodstream infections caused by extended-spectrum β-lactamase (ESBL)-producing *Enterobacterales* is increasing globally [1]. As ESBLs confer resistance to penicillins, cephalosporins and aztreonam, treatment options for infections due to these organisms are significantly restricted. In bloodstream infections caused by *Enterobacterales*, ESBL production is associated with an increase in the risk of delay in effective antimicrobial therapy, and a significantly higher mortality [2]. Moreover, the rise in prevalence of infections due to ESBL-producing *Enterobacterales* will inevitably lead to an increase in carbapenem usage, which will aggravate the global crisis of antimicrobial resistance.

CTX-M-type β-lactamases, which are postulated to originate from *Kluyvera* spp., account for the majority of ESBLs detected in clinical specimens [3]. Based on analysis of the amino acid sequences, CTX-M-type enzymes can be subclassified into multiple major clusters, namely CTX-M groups 1, 2, 8, 9 and 25. CTX-M-15 (group 1) is the most prevalent enzyme variant in Europe and America, whereas CTX-M-14 (group 9) is predominant in Asian countries such as China, South Korea and Japan [1]. In Hong Kong, more than 90% of ESBL-producing organisms produce CTX-M enzymes, with group 9 enzymes being the most prevalent followed by group 1 enzymes, which is consistent with other studies [4,5,6].

Detection of ESBLs in *Enterobacterales* with phenotypic methods such as double disc synergy tests and combination disc tests requires testing on isolated bacterial colonies. These tests require incubation which takes at least 16 to 20 h before results can be interpreted. In critical scenarios such as bloodstream infections, a long turnaround time may delay administration of effective antimicrobial therapy, which is associated with increased mortality. The NG-Test CTX-M MULTI (CTX-M LFA) test (NG-Biotech Laboratories, France) is a lateral flow assay that is able to detect CTX-M producers rapidly in positive blood culture (BC) fluids [7]. However, most of the previous studies on direct detection of CTX-M producers from positive BCs were carried out after bacterial identification confirmed the presence of *Enterobacterales* and only a few species were tested [7,8,9,10,11,12,13]. This study aimed to assess the diagnostic performance of the NG-CTX-M Test for direct detection of CTX-M producers in BCs positive for Gram-negative bacilli (GNB) before definitive identification.

## 2. Materials and Methods

### 2.1. Study Design and Microbiological Methods

In this study, we evaluated the performance of the NG-CTX-M test using both spiked blood cultures (BCs) with producers of CTX-M-type or CTX-M-like enzymes and clinical BCs. The NG-CTX-M test is a lateral flow immunochromatographic assay using anti-CTX-M mouse monoclonal antibodies for detection of groups 1, 2, 8, 9 and 25 CTX-M β-lactamases [7]. Testing involving spiked BCs were performed at the clinical microbiology laboratory at the Queen Mary Hospital (QMH-lab), which is a 1700-bed university-affiliated regional hospital in Hong Kong. Testing involving clinical BCs were performed at QMH-lab and another clinical microbiology laboratory in Kwong Wah Hospital (KWH-lab), which is a 1400-bed regional hospital in Hong Kong. The studies were conducted from April 2021 to March 2022.

In the two diagnostic laboratories, the BD BACTEC™ FX blood culture system was used for incubation of blood culture bottles. Blood culture bottles that were flagged positive after incubation were subjected to Gram staining and bacteria were recovered on appropriate agar media (Columbia blood agar, chocolate and MacConkey agars for aerobes or facultative anaerobes; neomycin blood agar supplemented with haemin and vitamin K for anaerobes) under 35 °C incubation in aerobic or anaerobic conditions as appropriate [6]. Microbial identification was performed on isolated bacterial colonies using matrix-assisted laser desorption/ionization-time of flight mass spectrometry (MALDI-TOF MS) with the Bruker Microflex^®^ LT Biotyper, according to the manufacturer’s instructions [14]. Antimicrobial susceptibility testing was performed using the disc diffusion method according to the Clinical and Laboratory Standards Institute [15]. For organisms identified to be members of the order *Enterobacterales*, phenotypic detection of ESBL production was performed using the combination disc test [15]. Discs with ceftazidime (30 μg), ceftazidime-clavulanate (30/10 μg), cefotaxime (30 μg) and cefotaxime-clavulanate (30/10 μg) were used. ESBL production was regarded as positive if there was a ≥5 mm increase in zone diameter for either antibiotic tested in combination with clavulanate, as compared with the agent when tested alone [15]. 

For testing of the spiked BCs, 42 well-characterized isolates which were producers of CTX-M (*n* = 15) or CTX-M-like (*n* = 17) enzymes were included. All the isolates were identified by MALDI-TOF MS and their susceptibility determined by the disc diffusion method. Bacterial strains were cultured overnight on a blood agar plate at 37 °C. For each isolate, an aerobic bottle was inoculated with a bacterial inoculum of ~1000 CFU in 1 mL sterile saline solution and incubated in the automated BACTEC system until flagged as positive. All the 42 strains were characterized by next-generation sequencing (NGS). 

For prospective testing of the clinical BCs, 350 clinical blood cultures positive for GNB collected between April 2021 and March 2022 from unique patients were included. In 130 clinical BCs, direct testing of the BC broth with the NG-test CTX-M MULTI was performed the next day after identification of the organism. Monobacterial BCs positive for members of the *Enterobacterales* order (*n* = 80) and other GNB (*n* = 50) were included in the study. Polymicrobial BCs or those positive for non-*Enterobacterales* GNB were excluded. These samples were included to allow investigation of a wide range of GNB. In another 220 consecutive, clinical BCs, direct testing with the CTX-M LFA was done immediately after the BC bottles were signaled positive and Gram staining revealed GNB, before further identification was carried out. Cultures with mixed GNB and non-GNB organisms (e.g., Gram positive organisms, fungi) were excluded. All the bacterial isolates from the clinical BCs were identified to species level, with susceptibility testing and detection of ESBL production as described above. In addition, the isolates were tested for the blaCTX-M gene using CTX-M group-specific multiplex polymerase chain reaction (PCR) assays [16].

### 2.2. Working Protocol for Directing Testing of Positive BCs with the NG-Test CTX-M MULTI

A procedure modified from the one described by Giordano et al. was used in which 150 µL of blood culture broth was mixed with 150 µL of NG-Test extraction buffer and incubated at 37 °C for 20 min to achieve better bacterial cell lysis [17]. Afterwards, 100 µL of the suspension was dispensed into the well of the immunochromatographic cassette. Results were read after 5 min and 15 min following the manufacturer’s recommendation [18]. A negative result was signified by the presence of only one line in the control region, whereas a positive result was indicated by the presence of two lines, in both the control and test regions.

### 2.3. Whole Genome Sequencing and Bioinformatics

Genomic DNA of each strain was extracted and qualified as was done in our previous studies [19,20]. All genomes were sequenced using the Illumina NovaSeq platform (Illumina, CA, USA) at the Novogene Bioinformatics Institute, Beijing. The workflow from raw reads to generate annotated genomes employed in our previous studies was performed [19,20]. Databases of CTX-M and CTX-M-like protein sequences were curated after comprehensively screening all available databases including the updated Bacterial Antimicrobial Resistance Reference Gene Database (https://www.ncbi.nlm.nih.gov/bioproject/PRJNA313047, accessed on 1 November 2022), BLDB (http://www.bldb.eu, accessed on 1 November 2022), ResFinder (upon the date of September 2022) and HMD-ARG-DB [21]. Predicted proteins from genomes were compared against CTX-M and CTX-M-like database with a 1 × 10^−4^ e-value using blastp. Proteins with identities higher than >60% and coverage higher than 70% were extracted and aligned with reference CTX-M-2 (CAA63263.1) using several methods including MAFFT, ClustalW and MUSCLE to validate the quality of alignment. The distance matrix was generated using MEGA11 [22]. 

### 2.4. Data Analysis

Data were presented as numbers and percentages (%). Results from the CTX-M LFA was compared against molecular detection of *bla*_CTX-M_ as the reference method. Sensitivity and specificity with 95% confidence interval (CI) were calculated using MedCalc software.

## 3. Results

### 3.1. Spiked Blood Cultures

A total of 42 well-characterized isolates were included in this study (Table 1). These isolates included producers of CTX-M-type (*n* = 15) or CTX-M-like enzymes (*n* = 28). The CTX-M-like enzymes included OXY-type (*n* = 5), FONA-type (*n* = 12), RAHN-type (*n* = 7), CumA-type (*n* = 1), SED-type (*n* = 1) and SFO-type (*n* = 1). Absence of CTX-M in the producers of CTX-M-like enzymes was verified by whole genome sequencing. Amino acid sequence alignment showed that the included CTX-M-like enzymes are closely related to the CTX-M-type enzyme at >60% protein pairwise identity (Figure 1 and Appendix A). In susceptibility testing, all of them were not susceptible to cefotaxime and/or ceftazidime and were ESBL-phenotype positive.

CTX-M β-lactamase detection was performed on positive aerobic blood culture bottles which were spiked with the isolates. The NG-Test CTX-M MULTI detected 100% of all CTX-M alleles from the M1 group (1, 15 and 55), M9 group (13, 14, 27, 65 and 191) and hybrid group (64, 123 and 132). Among the CTX-M-like enzymes, positive CTX-M result was obtained in 5 of 27 spiked bottles (Table 1). The isolates were producers of OXY-1-3 (*n* = 1), OXY-5-3 (*n* = 1), FONA-10 (*n* = 2) and SFO-1 (*n* = 1). The 28 isolates with CTX-M-like enzymes were further tested using colonies from culture growth on blood agar plates. Positive CTX-M results were obtained for the 5 isolates that yielded a positive result from the spiked blood cultures. In addition, a positive CTX-M result was obtained for another 7 isolates with OXY-2-8 (*n* = 1), FONA-8 (*n* = 1), FONA-9 (*n* = 2), FONA-11 (*n* = 1) and FONA-13 (*n* = 2) (Figure 2).

### 3.2. Clinical Blood Cultures

A total of 350 positive clinical BCs were tested. In 130 clinical BCs, direct testing was performed the next day after identification of the organisms was obtained. In 220 clinical BCs, direct testing was performed immediately after the bottles were flagged positive and before the organisms were identified (Table 2). In the BCs, 359 isolates of 46 different Gram-negative bacterial species were detected. The most common organisms were *Enterobacterales* (81.1%) (including *Escherichia coli* 56.3%, *Klebsiellla pneumoniae* 11.4%, *Proteus mirabilis* 5.4% and other *Enterobacterales* 8.0%), followed by *Pseudomonas* (8.2%) and anaerobic GNB (3.7%). Among the *Enterobacterales* isolates, 34.5% (98/284) were not susceptible to ceftazidime and/or ceftriaxone and 31.7% (90/284) exhibited the ESBL phenotype in the combined disc test. The number of ESBL producers tested in the retrospective and prospective clinical BCs were 38 and 52, respectively. In the prospective BCs, the prevalence of ESBL among *E. coli* and *Enterobacterales* were 30.9% (95% CI 23.9–39.1%) and 25.5% (95% CI 20.0–31.9%), respectively. During the study period, the monthly prevalence of ESBL in *Enterobacterales* ranged 13.5% to 35.0% without any increasing or decreasing trend. All the 90 ESBL-positive isolates (from 90 BCs) were PCR positive for the CTX-M gene. The number of positive results by the M1 and M9 group was 40 and 50, respectively. The other *Enterobacterales* isolates and those of other bacterial groups was PCR negative for the CTX-M gene (Table 2). The NG-Test MULTI assay yielded true-positive results in all the 90 monomicrobial samples with organisms that produce CTX-M. The organisms recovered from the samples included *E. coli* (*n* = 76), *K. pneumoniae*, (*n* = 9), *P. mirabilis* (*n* = 3), *K. varicola* (*n* = 1) and *Salmonella enterica* (*n* = 1). In the 260 BCs not containing CTX-M producers, the NG-Test MULTI assay yielded one false-positive result (monomicrobial) and 259 true-negative results (251 monomicrobial and 8 polymicrobial). A pure growth of *E. coli* was recovered from the only BC sample that yielded a false-positive NG-Test MULTI result. The organism was resistant to ampicillin and susceptible to cefuroxime, ceftriaxone, ceftazidime and meropenem. ESBL screening by the combined disc test was negative. The culture isolate was further tested using an NG-Test CTX-M MULTI assay and a negative result was obtained.

Considering all the clinical BCs together, the NG-Test MULTI assay showed 100% (95% CI, 96.0–100%) sensitivity and 99.6% (95% CI, 97.9–100%) specificity (Table 3). The sensitivity and specificity were similar for the BCs tested the next day and immediately after signaling positive. In the samples, the positive band was apparent at 5 min and no obvious difference was observed at 15 min.

## 4. Discussion

We present data on the diagnostic performance of CTX-M LFA for the detection of CTX-M producers directly on BC broths. The assay correctly detected all the CTX-M producers in spiked and clinical BCs. Previous studies showed that the CTX-M LFA could detect the 1, 2, 8, 9, 25 groups of CTX-M β-lactamases [7]. We further show that it could detect the chimeric hybrid CTX-M variants which have evolved by homologous recombination of two different CTX-M genes [23,24]. A novel finding is that several variants of the OXY, FONA and SFO β-lactamases were found to cross-react with antibodies in the CTX-M LFA, giving rise to false-positive results. It is intriguing that no correlation was found between cross-reactivity and pairwise identity with CTX-M-2, which is their closest CTX-M member in phylogenetic analysis. In the case of OXY β-lactamases, it was previously hypothesized that residues of the Lys-Lys-Ser (KKS) triplet at Amber positions 101 to 103 was responsible for the cross-reactivity and explains why false-positive results occurred with the OXY-1 group but not the OXY-2 group variants [25]. Our results showed that several β-lactamases with amino acid triplets other than KKS at those positions also yielded false-positive results, including an OXY-2 variant and several other β-lactamases (FONA-8, -9, -11, -13 and SFO-1). This indicates that cross reactivity is complex and might be influenced by protein folding and the level of β-lactamase expression. OXY and FONA are chromosomally-encoded ESBLs in *K. oxytoca* and *S. fonticola*, respectively, whereas SFO-1 is an uncommon plasmid-mediated ESBL that has been detected in members of *Enterobacterales* [25,26,27]. Since these are ESBL-producers anyway, the cross-reactions with CTX-M are of little clinical relevance. 

This study used a simple method to prepare the BC broths before allowing for migration on the cassette. After mixing the BC broth with the extraction buffer, we incubated the mixture at 37 °C for 20 min to improve the bacterial lysis. This incubation step was not included in the simplified protocol described by Giordano et al. [17]. In our pilot testing, we found that false-negative or weak positive results may occur with some CTX-M producers in BC broths and that the issue can be resolved by adding this incubation step. In all previous studies that evaluate CTX-M LFA, testing was performed on cell pellets obtained from BC broths, thus involving centrifugation, removal of supernatant and resuspension [7,8,9,10,11,12,13]. Furthermore, most previous studies performed CTX-M LFA after bacterial identification was obtained using the cell pellet from BC broth (Table 4). Only one study involving 100 BCs performed CTX-M LFA following Gram staining and before bacterial identification [7]. However, details of the GNB in the BCs were not described [7]. Although identification of organisms in BC broths may be achieved by MALDI-TOF MS, an earlier step of bacterial extraction is mandatory [28]. As the steps for obtaining the cell pellets from BC broth are labor-intensive, this may affect the cost-benefit ratio of performing the CTX-M LFA on a few species after identification. In our locality, MALDI-TOF MS-based identification of pathogens directly from positive BCs has not been widely adopted [29]. 

With the use of a simple extraction step, the CTX-M LFA identified 100% of *Enterobacterales* with the ESBL phenotype from BC broths. This is because 100% of the ESBL-positive isolates were CTX-M producers. All CTX-M producers were reliably detected in our study whereas previous studies have described a small number of false-negative results, which may be attributed to organisms with a mucoid colony morphology or low level CTX-M expression (Table 4) [10,13]. We encountered one instance of false-positive CTX-M LFA result involving BC positive for a drug-sensitive *E. coli*. In the BC sample, the false-positive result was also obtained using an alternative extraction method and the findings were published [18]. Walter et al. performed direct CTX-M LFA on cell pellets from 167 spiked BC broths, and two CTX-M-negative *Enterobacterales* isolates produced a weak band, which may be variably interpreted by different observers as false-positive or invalid results [13]. In the present study, no weak band for the CTX-M producers was encountered. The sensitivity and specificity from our evaluation are consistent with those reported in previous studies (Table 4). Nonetheless, we must note that direct testing of BC using CTX-M LFA is only appropriate in countries where CTX-M is prevalent. In areas where non-CTX-M-type ESBLs are common, hydrolysis-based assays could be more suitable [9].

This study has some limitations. The diagnostic performance was based on high CTX-M prevalence in local settings and it may not be generalizable to other countries. As the CTX-M LFA only targets one resistance mechanism, a negative result cannot be used to infer extended-spectrum cephalosporin susceptibility. Moreover, the impact of rapid detection of CTX-M on antibiotic use and patient outcomes was not assessed. The simple workflow, evaluations in two hospital laboratories, the inclusion of a wide range of BC positive for GNB other than *Enterobacterales* and testing of consecutively positive clinical BC samples are the present study’s strengths. In many countries, carbapenemase-producing *Enterobacterales* is emerging and there may also be a need for early detection in blood cultures. Studies have showed that LFA can also be used for rapid detection of five common types of carbapenemases in positive blood cultures [11]. 

## 5. Conclusions

This study showed that rapid detection of CTX-M producers in BC broths can be reliably achieved using the CTX-M LFA. The simple BC preparation method offers an opportunity for implementation in clinical laboratories for early optimization of antibiotics.

## Figures and Tables

**Figure 1 microorganisms-11-00128-f001:**
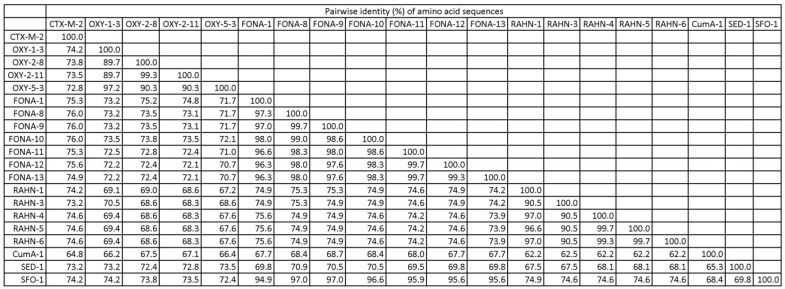
Distance matrix of amino acid sequences of CTX-M-2 and CTX-M-like β-lactamases.

**Figure 2 microorganisms-11-00128-f002:**
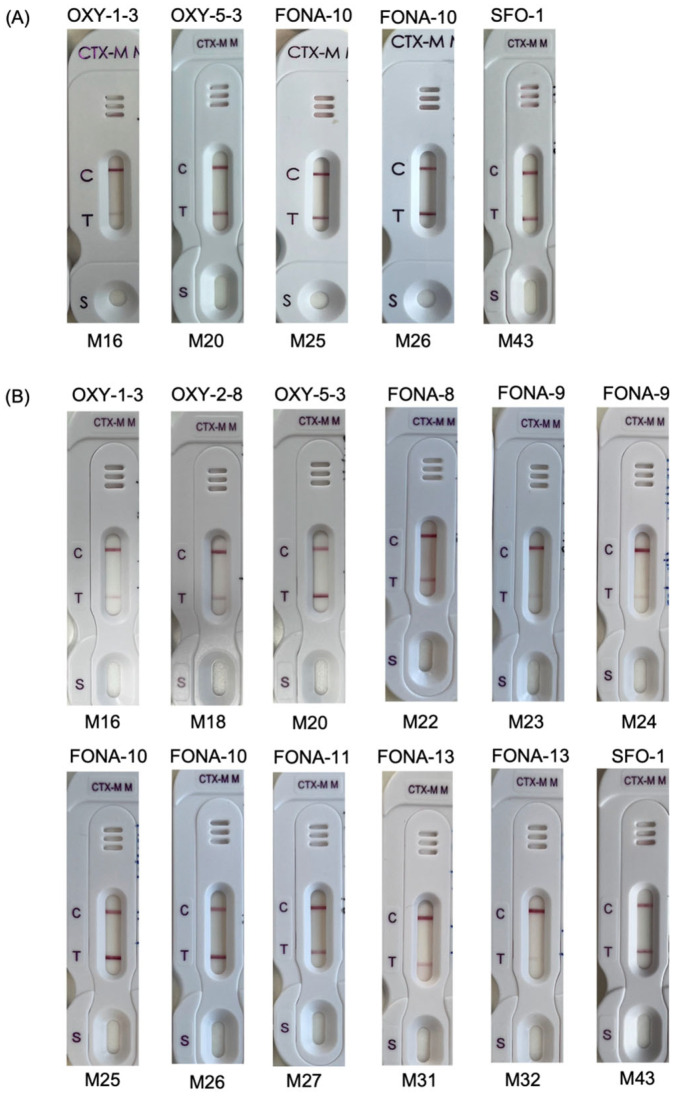
NG-Test CTX-M MULTI results obtained for (**A**) positive blood culture spikes with 5 isolates expressing CTX-M-like β-lactamase; (**B**) bacterial colonies of 12 isolates expressing CTX-M-like β-lactamase. The labels above and below each photo show the β-lactamase name and sample name, respectively.

**Table 1 microorganisms-11-00128-t001:** CTX-M MULTI results of reference isolates from spike blood cultures and culture growth.

Group	Organism	ESBL	*n*	Spiked Blood Culture (No.)		Culture Growth (No.)	
				Positive	Negative	Positive	Negative
CTX-M-1G	*E. coli*	CTX-M-1	2	2			
	*E. coli*	CTX-M-15	2	2			
	*E. coli*	CTX-M-55	1	1			
CTX-M-9G	*E. coli*	CTX-M-13	1	1			
	*E. coli*	CTX-M-14	1	1			
	*E. coli*	CTX-M-27	1	1			
	*E. coli*	CTX-M-65	1	1			
	*E. coli*	CTX-M-191	1	1			
CTX-M hybrid	*E. coli*	CTX-M-64	1	1			
	*E. coli*	CTX-M-123	2	2			
	*E. coli*	CTX-M-132	2	2			
OXY	*K. michiganensis*	OXY-1-3	1	1		1	
	*K. oxytoca*	OXY-2-8	2		2	1	1
	*K. oxytoca*	OXY-2-11	1		1		1
	*K. oxytoca*	OXY-5-3	1	1		1	
FONA	*S. fonticola*	FONA-1	1		1		1
	*S. fonticola*	FONA-8	1		1	1	
	*S. fonticola*	FONA-9	2		2	2	
	*S. fonticola*	FONA-10	2	2		2	
	*S. fonticola*	FONA-11	2		2	1	1
	*S. fonticola*	FONA-12	2		2		2
	*S. fonticola*	FONA-13	2		2	2	
RAHN	*R. aquatilis*	RAHN-1	2		2		2
	*R. aquatilis*	RAHN-3	1		1		1
	*R. aquatilis*	RAHN-4	1		1		1
	*R. aquatilis*	RAHN-5	1		1		1
	*R. aquatilis*	RAHN-6	2		2		2
Others	*P. vulgarus*	CumA-1	1		1		1
	*C. sedlakii*	SED-1	1		1		1
	*K. aerogenes*	SFO-1	1	1		1	

ESBL, extended-spectrum β-lactamase.

**Table 2 microorganisms-11-00128-t002:** Summary of positive clinical blood culture bottles tested in this study.

	No. (%) of Clinical Blood Cultures	No. of Species in the Blood Cultures
Hospital source		
KWH-lab	180 (51.4)	26
QMH-lab	170 (48.6)	29
Sample origin		
Aerobic bottle	178 (50.9)	27
Anaerobic bottle	172 (49.1)	28
Detected organisms		
*Escherichia coli*	197 (56.3)	1
*Klebsiella pneumoniae*	40 (11.4)	1
*Proteus mirabilis*	19 (5.4)	1
Other *Enterobacterales* ^a^	28 (8.0)	16
*Pseudomonas*	29 ((8.2)	6
Anaerobes	13 (3.7)	8
Acinetobacter	9 (2.6)	3
Other bacteria ^a^	15 (4.3)	10
Number of organisms		
Monomicrobial	342 (97.7)	42
Polymicrobial	8 (2.3)	11
CTX-M identified		
CTX-M M1 group	40 (11.4)	4
CTX-M M9 group	50 (14.3)	3
Negative for CTX-M	260 (74.3)	45
Total	350 (100)	46

^a^ See Appendix A for the full list of organisms.

**Table 3 microorganisms-11-00128-t003:** Summary of test results from clinical blood cultures.

Sample Description	Result (No.)				Test Performance ^a^	
	TP	FP	FN	TN	Sensitivity	Specificity
Testing performed the next day on 130 clinical BCs following bacterial identification	38	0	0	92	100% (91.0–100%)	100% (96.1–100%)
Testing performed on 220 clinical BCs immediately after Gram stain showing Gram negative bacilli	52	1	0	167	100% (93.1–100%)	99.4% (96.7–100%)
Total	90	1	0	259	100 (96.0–100%)	99.6% (97.9–100%)

TP, true-positive; FP, false-positive; FN, false-negative; TN, true-negative. ^a^ 95% confidence interval given inside brackets.

**Table 4 microorganisms-11-00128-t004:** Published literature on direct testing of positive blood cultures using an NG-Test CTX-M MULTI assay.

Source	Year	No. of Clinical (Spike) BCs Tested	Organisms in BCs (No. of Species)	Reference Method ^g^	No.				Sensitivity, % (95% CI)	Specificity, % (95% CI)
					TP	TN	FP	FN		
[8]	2020	166	*Enterobacterales* (11) ^a^	G	43	123	0	0	100 (91.8–100)	100 (97.1–100)
[7]	2020	100	Gram negative bacilli ^b^	G	10	90	0	0	100 (69.2–100)	100 (96.1–100)
[12]	2021	49 (32)	*Enterobacterales* (6) ^c^	P	26	55	0	0	100 (86.8–100)	100 (93.5–100)
[13]	2021	0 (167)	*Enterobacterales* (9) ^d^	G	124	38	2	3 ^h^	97.6 (93.3–99.5)	95.0 (83.1–99.4)
[10]	2022	61 (19)	*Enterobacterales* (3) ^e^	P	26	53	0	1 ^h^	96.3 (81.0–99.9)	100 (93.3–100)
[9]	2022	142	*Enterobacterales* (2) ^f^	P	34	105	0	3 ^i^	91.9 (78.1–98.3)	100 (96.6–100)
[11]	2022	1055	*Enterobacterales* (2) ^f^	P	273	757	0	25	91.6 (87.9–94.5)	100 (99.5–100)

CI, confidence interval; TP, true-positive; TN, true-negative; FP, false-positive; FN, false-negative. ^a^ Including *Citrobacter koseri*, *Enterobacter aerogenes*, *E. cloacae*, *Escherichia coli*, *Hafnia alvei*, *Klebsiella oxytoca*, *K. pneumoniae*, *Morganella morganii*, *Pantoea septica*, *Proteus mirabilis*, *Serratia marcenscens*. ^b^ Details of the species for the 100 Gram negative bacilli not reported. ^c^ Including *E. cloacae*, *E. coli*, *Klebsiella* spp., *P. mirabilis*, *Salmonella* spp., *S. marcescens*. ^d^ Including *C. freundii*, *E. cloacae*, *E. coli*, *K. aerogenes*, *K. pneumoniae*, *M. morganii*, *P. mirabilis*, *S. marcescens*, *S. typhimurium*. ^e^ Including *E. coli*, *K. oxytoca*, *K. pneumoniae*. ^f^ Including *E. coli*, *K. pneumoniae*. ^g^ G, *bla*_CTX-M_ gene detection; P, ESBL phenotype. ^h^ The isolates were *bla*_CTX-M_ positive. ^i^ The isolates were *bla*_CTX-M_ negative.

## Data Availability

All relevant data have been included in this manuscript.

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
