# Peer review of "Evaluation of a Lateral Flow Immunoassay for Rapid Detection of CTX-M Producers from Blood Cultures"

_microorganisms, 2023, doi:10.3390/microorganisms11010128_

Round 1
Reviewer 1 Report
I read with interest the paper by Fang et al. They evaluated the NG-test CTX-M MULTI immunoassay directly from spiked and clinical Gram-negative positive blood cultures.
The topic is of interest despite the fact that several papers have already investigated the performance of NG-test CTX-M assay and more on its clinical impact are currently being published.
In this regard, I am not sure how much of a strength the study can be having performed the test in several cases without knowing the bacterial identification of the isolate (see lines 54-59). Studies already published have used the test on a few bacterial species after identification precisely to maximise the cost-benefit ratio.
The results found by the authors on cross-reactions are certainly interesting but from a clinical point of view they are of little relevance since all isolates were ESBL-producing anyway.
From a clinical point of view, it would be interesting to know whether the authors have found carbapenemase-producing isolates co-expressing CTX-M and what kind of diagnostics they envisage in such cases.
Author Response
I read with interest the paper by Fang et al. They evaluated the NG-test CTX-M MULTI immunoassay directly from spiked and clinical Gram-negative positive blood cultures.
The topic is of interest despite the fact that several papers have already investigated the performance of NG-test CTX-M assay and more on its clinical impact are currently being published.
RESPONSE
Thank you for the useful input for us to improve the manuscript.
In this regard, I am not sure how much of a strength the study can be having performed the test in several cases without knowing the bacterial identification of the isolate (see lines 54-59). Studies already published have used the test on a few bacterial species after identification precisely to maximise the cost-benefit ratio.
RESPONSE
Although MALDI-TOF can be used for bacterial identification in positive BCs, this requires obtaining the cell pellets from BC broth and involving centrifugation, removal of supernatant and resuspension. The labor cost would also need to be considered in assessing the cost-benefit ratio. In countries where direct bacterial identification is performed, performing the CTX-M LFA without knowing the bacterial identification would still provide clinically relevant information for optimizing antibiotic use. We revised the discussion part to address this comment.
The results found by the authors on cross-reactions are certainly interesting but from a clinical point of view they are of little relevance since all isolates were ESBL-producing anyway.
RESPONSE
We added this point to the end of the first paragraph, Discussion section
From a clinical point of view, it would be interesting to know whether the authors have found carbapenemase-producing isolates co-expressing CTX-M and what kind of diagnostics they envisage in such cases.
RESPONSE
Thank you for the suggestion. We a short discussion on this in the last paragraph, Discussion section.

Reviewer 2 Report
I have read with interest the manuscript submitted by Fang et al.
I have some comments to be addressed in order to improve the quality of this manuscript:
Please provide more information regarding the retrospective BCs.
Please consider adding some chronological information regarding the type of ESBL produced during the studied period, if it was constant or if a specific type of CTX-M ESBL decreased/increased. It would be interesting to know also if the incidence varied during the studied period.
Table 4 - legend, please use italics when writing the bacteria names.
row 282 - replace GNR with GNB.
Please notice that I cannot access the supplementary data listed on zenovo (
Permission required
You do not have sufficient permissions to view this page.)
Author Response
I have read with interest the manuscript submitted by Fang et al.
I have some comments to be addressed in order to improve the quality of this manuscript:
Please provide more information regarding the retrospective BCs.
RESPONSE
The number of ESBL producers tested in the retrospective BCs has been added to the result. Details of the organisms tested in the retrospective BCs was given in supplementary file, Table S1.
Please consider adding some chronological information regarding the type of ESBL produced during the studied period, if it was constant or if a specific type of CTX-M ESBL decreased/increased. It would be interesting to know also if the incidence varied during the studied period.
RESPONSE
The CTX-M was only categorized to subgroups using PCR assays. The allele of the CTX-M type was not determined. Hence, changes in CTX-M allele cannot be analyzed.
During the study period, the monthly prevalence of ESBL in Enterobacterales ranged 13.5% to 35.0%. Since the number of isolates per month was relatively small and the study period is relatively short, no conclusion can be drawn on the trend. This has been added to the results in the revision.
Table 4 - legend, please use italics when writing the bacteria names.
RESPONSE
The names have been italicized
row 282 - replace GNR with GNB.
RESPONSE
This has been revised
Please notice that I cannot access the supplementary data listed on zenovo
RESPONSE
The supplementary data is appended here for information by this reviewer.
Table S1. List of organisms detected in clinical blood cultures in this study
Figure S1. Amino acid sequence alignment of CTX-M-2 with CTX-M like -lactamases investigated in this study
